# *Lactiplantibacillus argentoratensis* and *Candida tropicalis* Isolated from the Gastrointestinal Tract of Fish Exhibited Inhibitory Effects against Pathogenic Bacteria of Nile Tilapia

**DOI:** 10.3390/vetsci10020129

**Published:** 2023-02-07

**Authors:** Noppadon Siangpro, Songkran Chuakrut, Wanna Sirimanapong, Somboon Tanasupawat, Wongsakorn Phongsopitanun, Bunyarit Meksiriporn, Jarungwit Boonnorat, Siripun Sarin, Siriwat Kucharoenphaibul, Rumpa Jutakanoke

**Affiliations:** 1Department of Microbiology and Parasitology, Faculty of Medical Science, Naresuan University, Phitsanulok 65000, Thailand; 2Department of Clinical Sciences and Public Health, Faculty of Veterinary Science, Mahidol University, Nakhon Pathom 73110, Thailand; 3Department of Biochemistry and Microbiology, Faculty of Pharmaceutical Sciences, Chulalongkorn University, Bangkok 10330, Thailand; 4Department of Biology, Faculty of Science, King Mongkut’s Institute of Technology Ladkrabang, Bangkok 10520, Thailand; 5Department of Environmental Engineering, Faculty of Engineering, Rajamangala University of Technology Thanyaburi (RMUTT), Klong 6, Pathum Thani 12110, Thailand; 6Center of Excellence in Fungal Research, Faculty of Medical Science, Naresuan University, Phitsanulok 65000, Thailand

**Keywords:** probiotic, Nile tilapia, pathogenic bacteria, lactic acid bacteria, yeast

## Abstract

**Simple Summary:**

The two lead probiotic isolates (AT8/5 and YON3/2) demonstrating some probiotic characteristics were further identified by phylogenetic identification. AT8/5 and YON3/2 showed the highest similarity to *Lactiplantibacillus argentoratensis* and *Candida tropicalis*, respectively. These isolates are potential functional candidates for use as probiotics in aquaculture.

**Abstract:**

Nile tilapia is one of the most consumed farmed fish in the world. The outbreak of pathogenic bacterial diseases causes high mortality rates and economic losses in Nile tilapia farming. Antibiotic administrations are commonly utilized to inhibit and prevent bacterial infections. However, antibiotics are expensive and cause serious concerns for antibiotic resistance in fish that can be potentially transferred to humans. As an alternative solution, probiotics can be used to prevent infection of pathogenic bacteria in fish. In this work, both bacteria and yeast were isolated from fish gastrointestinal tracts and their inhibitory activity against Nile tilapia pathogenic bacteria was evaluated, as well as other probiotic properties. In this study, 66 bacteria and 176 acid tolerant yeasts were isolated from fish gastrointestinal tracts. Of all isolated microorganisms, 39 bacterial and 15 yeast isolates with inhibitory effect against pathogens were then examined for their probiotic properties (acidic and bile salt resistance, adhesion potential, and biofilm formation), formation of antibacterial factor survival rate under simulated gastrointestinal fluid, and safety evaluation. AT8/5 bacterial isolate demonstrated probiotic properties and the highest inhibition against all 54 tested pathogens while YON3/2 yeast isolate outperformed the inhibitory effect among all yeast isolates. These two probiotic isolates were further identified by 16S rDNA and the D1/D2 domain of 26S rDNA sequence analysis for bacterial and yeast identification, respectively. AT8/5 and YON3/2 showed the highest similarity to *Lactiplantibacillus argentoratensis* and *Candida tropicalis*, respectively. This is the first report on isolated *L. argentoratensis* and *C. tropicalis* with antipathogenic bacteria of Nile tilapia properties. Collectively, AT8/5 and YON3/2 could be potentially used as promising alternatives to existing antibiotic methods to prevent pathogenic bacteria infection in Nile tilapia farming.

## 1. Introduction

Nile tilapia (*Oreochromis niloticus*) has been recognized as one of the most economically important freshwater aquaculture species in the last three decades, with a global market value of USD 7.9 billion in 2020 [1]. Disease outbreaks have been regarded as the major obstacle to Nile tilapia farming worldwide [1]. Fish pathogenic bacteria, especially *Streptococcus agalactiae,* outbreaks cause high mortality rates in Nile tilapia farming with losses being reported up to 70% [2]. The most common approach to inhibit and prevent pathogenic bacterial infections has been antibiotic administration. However, antibiotic usage and misuse leads to major concerns regarding antibiotic side effects. Misused antibiotics have stimulated a selection of antibiotic-resistant bacteria, escalating zoonotic risk in human health [3]. Antibiotic residues in fish can also hamper international trade because most countries refuse to import of antibiotic-treated fish products [4,5]. To address the shortcomings due to antibiotic usage, probiotics are one promising alternative to prevent pathogenic bacterial infection in fish.

Probiotics are live microorganisms that can confer health benefits and protective immunity on hosts when administered in proper amounts [6]. Probiotics possess several protective mechanisms against bacterial infections, including antimicrobial compound production, reduction of serum cholesterol, improvement of lactose metabolism, and prevention of colonization of pathogens by competitive inhibition. Probiotics can also stimulate mucosal barrier function and influence various aspects of the innate and acquired immune system by inducing phagocytosis and IgA secretion, modifying T lymphocyte response, enhancing T helper 1 response, and attenuating T helper 2 response [7]. Additionally, probiotics can lower gut pH, release gut protective metabolites, and regulate intestinal motility and mucus production [8].

Probiotic bacteria such as lactic acid bacteria (LAB), *Bifidobacterium*, and *Bacillus* spp. are recognized as the most common probiotic microorganisms. Several studies on the use of probiotic bacteria in aquaculture have indicated the capability for rendering protection against pathogenic bacteria infection, enhancing growth performance, and improving immune system in fish [9]. Additionally, various studies reported that the screening and selection of probiotic bacteria from fish gastrointestinal (GI) tracts such as Nile tilapia [9,10] and Climbing perch (*Anabas testudineus*) [11] were successfully performed because fish GI tracts contain diversified, significant, and beneficial microbes that could be potential probiotics. Therefore, isolating probiotic microbes from fish GI tracts was an interesting approach to isolate probiotics against pathogenic infection in fish including Nile tilapia.

In addition to bacterial probiotics, yeasts have increasingly been regarded as potential candidates for probiotics. In the last few years, there has been an increase in research involving screening for novel yeast with probiotic properties. Several studies have demonstrated that many yeasts possess an ability to enhance growth performance, stimulate immune response, as well as protect and reduce mortality from bacterial infection in many fish species [12]. Probiotic yeast strains have been isolated from varied sources such as fruit, fermented foods, dairy products, and fish GI tracts. Various studies have shown that yeasts isolated from GI tracts of fish, such as Mullet (*Mugil* spp.) [13], Salmonids, Corvina drum (*Cilus gilberti*), and Great amberjack (*Seriola lalandi*) [12], demonstrated potential probiotic properties. In spite of their beneficial probiotic properties, isolation and screening of probiotic properties of yeasts have been largely focused on a few species [14]. There have been few reports on the isolation of probiotic yeast research from GI tracts compared to probiotic bacterial isolation [15].

Bacteria and yeast constitute a dominant part of the gut microbiome of fish [14,16]. The potential of these gut-isolated bacteria and yeast to display probiotic properties and antipathogenic bacterial infection represents a promising alternative for their potential application in developing of feed additives for Nile tilapia farming. However, the isolation of both probiotic bacteria and yeast from fish GI tracts has not been reported. In this work, we isolated both bacteria and yeast from fish GI tracts and evaluated their probiotic properties and inhibitory activities against Nile tilapia’s pathogenic bacteria. Following validation, our gut-isolated bacteria and yeasts demonstrated probiotic properties and antibacterial ability in different fish species. Hence, our probiotic bacteria and yeast uncovered from fish GI tracts represent a viable alternative for an application used in Nile tilapia fish farming and are transferable to other fish species to prevent the outbreak of pathogenic bacterial infection.

## 2. Materials and Methods

### 2.1. Sample Collection

Samples of fish GI tracts were collected from fish shops in markets in Muang Phitsanulok, Phitsanulok, Thailand and transported to a laboratory within 45 min using an ice box. In total, 27 samples were collected, included 15 Climbing perch, two Striped snakehead (*Channa striata*), one Soldier river barb (*Cyclocheilichthys enoplos*), one Common carp (*Cyprinus carpio*), three Asian sea bass (*Lates calcarifer*), and five Nile tilapia (*O. niloticus*). 

### 2.2. Isolation of Probiotic Microorganisms

Under sterile conditions, the samples were homogenized with 30 mL of 0.85% (*w*/*v*) NaCl using a stomacher (IUL Instrument). The homogenates were transferred into Erlenmeyer flasks containing 50 mL of Yeast Malt (YM) broth (HiMedia) and Lactobacillus MRS broth (HiMedia). After inoculation, MRS broth was incubated under limited oxygen conditions at 30 °C for 48 h, while YM broth flasks were incubated under aerobic conditions at 30 °C and 200 rpm shaking for 48 h. Each homogenate was diluted and spread onto petri dishes containing Lactobacillus MRS agar (HiMedia) plus 0.004% (*w*/*v*) bromocresol purple and selective medium YM agar (HiMedia) pH 2.5 for MRS broth and YM broth homogenate, respectively. Plates were incubated under the oxygen-limited condition at 30 °C for 24–48 h (for MRS agar) and aerobic condition at 30 °C for 48 h (for YM agar pH 2.5). For bacteria, individual colonies with a yellow diffusion halo were selected and purified by a cross-streaking method in a fresh MRS agar. For yeast, individual colonies were selected and purified by cross streaking method in a fresh YM agar pH 2.5. Cell morphology and arrangement of the pure isolated bacteria and yeasts were observed using the Gram staining (M&P IMPEX) and wet mount (Fluka) method, respectively. All the isolates were stored at −20 and −80 °C in MRS and YM broth supplemented with 30% (*v*/*v*) glycerol for bacteria and yeasts, respectively.

### 2.3. Pathogenic Bacterial Strains Used

Six fish pathogenic bacterial strains were selected for the present study, including *Aeromonas hydrophila* MUVS 2018 AH 001, *Aeromonas sobria* MUVS 2017 AS, *Aeromonas veronii* vsmu 083, *Edwardsiella ictaluri* 2010/12 EI, *Edwardsiella tarda* MUVS 2018 ET1, and *Streptococcus agalactiae* MUVS 2017 SA 001. These were isolated from infected Nile tilapia received from Assistant Professor Wanna Sirimanapong, Department of Clinical Sciences and Public Health, Faculty of Veterinary Science, Mahidol University.

### 2.4. Pathogenic Bacteria Inhibiting Test

#### 2.4.1. Probiotic Bacterial Isolate Preparation

The suspension of bacterial isolates at a density of 0.5 on the McFarland standard (~1.5 × 10^8^ CFU/mL) by using a McFarland densitometer (Biosan) was swabbed onto MRS agar and incubated under limited oxygen conditions at 30 °C for 24 h. 

#### 2.4.2. Probiotic Yeast Isolates Preparation

The suspension of yeast isolates at a density of 2.0 on the McFarland standard (~6.0 × 10^8^ CFU/mL) was swabbed onto YM agar and incubated under aerobic conditions at 30 °C for 48 h.

#### 2.4.3. Inhibitory Capacity against Fish Pathogenic Bacteria

The analysis of pathogenic bacteria inhibiting activity of probiotic isolates was performed with the agar slab method described by Yamashita et al. [9] with slight modifications. The pathogenic bacteria strain grown in 5 mL Tryptone Soya broth (TSB) (HiMedia) was adjusted to 0.5 McFarland standard and then swabbed onto Müeller–Hinton agar (MHA) (HiMedia). After that, 6 mm slabs were cut out from the probiotic MRS and YM plates and then transferred onto plates prepared with MHA swabbed with pathogenic bacteria strains. The plates were incubated at 30 °C for 24 h under limited oxygen (for probiotic bacteria) and aerobic conditions (for probiotic yeast) for 48 h. The diameters of the inhibition zone were then measured using a vernier caliper. MRS and YM agar without microbial growth served as the negative controls, while antibiotic discs were used as positive controls i.e., 30 µg/disc tetracycline for *Aeromonas* spp. and *S. agalactiae* and 30 µg/disc chloramphenicol for *Edwardsiella* spp.

### 2.5. Probiotic Properties Tests

#### 2.5.1. Bile Salt Tolerance Test

The ability of probiotics to survive in the presence of bile salt was analyzed by the method described by Khagwal et al. [17] with minor modifications. The 24 h and 48 h cultures of the probiotic bacteria and yeast showing pathogenic bacteria inhibition, respectively, were adjusted to 0.5 (bacteria) and 2.0 (yeast) McFarland standard and then centrifuged at 4000× *g* for 10 min. Afterward, the pellets were resuspended in sterile 0.85% (*w*/*v*) NaCl. Then, 100 µL of cell suspension was inoculated in 5 mL MRS and YM broth containing 1% (*w*/*v*) bile salt [18] for bacteria and yeast, respectively, and incubated at 30 °C under limited oxygen (for bacteria) and aerobic conditions (for yeast). Growth was measured by taking optical density (OD) after a time interval of 6 and 24 h at 640 nm using a spectrophotometer (Metertech). The OD_640_ of cells in the presence of bile salt was compared to the OD_640_ of cells inoculated without bile salt. An increase in OD_640_ indicated probiotic growth.

#### 2.5.2. Acid Tolerance Test

The resistance of probiotic bacteria to low pH was examined by the method described by Khagwal et al. [17] with slight modifications. The fresh overnight culture of the probiotic bacteria showing pathogenic bacteria inhibition was adjusted to 0.5 McFarland standard and then centrifuged at 4000× *g* for 10 min and resuspended in sterile 0.85% (*w*/*v*) NaCl. Then, 100 µL of cell suspension was inoculated in 5 mL MRS broth adjusted with 1N HCl to pH 2.5. The inoculated broth was incubated at 30 °C under limited oxygen conditions. After 6 and 24 h of incubation, the growth of the probiotic bacteria was measured using a spectrophotometer and OD_640_ was compared to the control group inoculated in MRS broth pH 6.5.

#### 2.5.3. Adhesion Property Test

The ability of the probiotics to adhere to abiotic surfaces (glass) was evaluated as recommended by Yegorenkova et al. [19] with slight modifications. Probiotic bacteria cultures were grown for 24 h in the MRS broth and adjusted to 0.5 McFarland standard. Probiotic yeast cultures were grown for 24 h in YM broth, and then adjusted to 2.0 McFarland standard. Both probiotic bacteria and yeast cultures were diluted 100-fold with sterile MRS or YM broth and inoculated into 16 × 100 mm borosilicate glass test tubes containing MRS or YM broth (200 µL per test tube). The control test tubes received only the broth media. The probiotic bacteria cultures were then incubated unshaken at 30 °C for 24 h, while probiotic yeast cultures were incubated at 30 °C, 200 rpm with shaking for 48 h. After incubation, the cultures were carefully withdrawn, and the test tubes were washed twice with distilled water to remove non-adherent cells. Then, 5 mL of 2% (*w*/*v*) crystal violet was added to each tube to stain attached cells for 15 min at room temperature. Dye solution was then discarded. Test tubes were carefully washed twice with distilled water. Dye that had bound to the cells adsorbed on the wall of the test tubes was dissolved into 5 mL of 95% (*v*/*v*) ethanol. The adherent ability was evaluated by the absorbance at 590 nm (Abs_590_) measured by a spectrophotometer.

#### 2.5.4. Biofilm Formation Test

The method used for screening probiotic isolates for biofilm formation was Congo red agar (CRA) method as described by Freeman et al. [20] and Mahdhi et al. [21] with slight modifications. The medium was composed of brain heart infusion (BHI) broth (HiMedia) 37 g/L, sucrose 50 g/L, agar 15 g/L and Congo red 0.8 g/L. Selected probiotics were incubated in media under aerobic conditions for 24–48 h at 30 °C for yeast or limited oxygen for 24 h at 30 °C for bacteria. Black colony in the black medium indicated a positive result while the non-biofilm producer was expected to yield red colony and red medium.

### 2.6. Detection of Antibacterial Factor Secretion of Selected Probiotic Isolates

The antibacterial factor secretion was also detected following the procedure as described by Schillinger and Lücke [22] with slight modifications. Selected probiotic bacteria isolates were grown in MRS broth under limited oxygen conditions at 30 °C for 24 h. Probiotic yeast were aerobically grown in YM broth at 30 °C for 48 h. Cell-free supernatant (CFS) of each isolate was obtained by centrifugation at 15,000× *g* for 15 min at 4 °C. The supernatant of probiotic bacteria was divided into two groups. The first group was adjusted to pH 7.0 with 1 N NaOH and subsequently boiled for 3 min at 100 °C to eliminate acid compounds and heat labile bacteriocin substrates (Neutralized CFS) while the second group received neither pH adjustment nor boiling (Non-neutralized CFS). The CFS of yeast isolated was prepared into two groups in a similar fashion. Antibacterial activity was determined using an agar well diffusion assay. MHA was swabbed with 0.5 McFarland cell density of each fish pathogenic bacteria strain. Then, 30 µL of neutralized and non-neutralized CFS was added into each well (6 mm diameter). The plates were then incubated at 30 °C for 24 h and inhibition zone was measured by using vernier caliper.

### 2.7. Simulated GI Tract Tolerance Test

Conditions in the stomach and intestine (GI tract) were simulated using the method described by Pennacchia et al. [23] with slight modifications. Each selected probiotic bacterium was grown in 5 mL of MRS broth under limited oxygen conditions at 30 °C for 24 h. Each selected probiotic yeast was grown in 5 mL of YM broth under aerobic conditions at 30 °C, 200 rpm shaking, for 48 h. The cultures were adjusted to 0.5 and 2.0 McFarland standards for bacteria and yeast, respectively. Then, 1 mL of density-adjusted cultures was transferred to sterile microcentrifuge tubes and centrifuged at 7000× *g* for 5 min at 4 °C. The pellet was washed with 0.85% *w*/*v* NaCl and resuspended in simulated gastric juice containing: 8.58 g/L phosphate buffer saline (PBS), 3 g/L pepsin, adjusted the pH to 2.5 with 1 N HCl, and then incubated at 30 °C for 3 h under limited oxygen conditions. Simulated gastric juice tolerance was examined by determining total viable cell counts in gastric juice withdrawn at 0, 1.5, and 3 h by standard plate count method on MRS and YM agar for bacteria and yeast, respectively. After 3 h of incubation in simulated gastric juice, the suspension was centrifuged at 7000× *g* for 5 min at 4 °C, and the pellet was washed by 0.85% *w*/*v* NaCl and resuspended with simulated intestinal fluid containing 8.58 g PBS, 10 g bile salt, and 1 g pancreatin per liter of distilled water. The pH was adjusted to 2.5 with NaOH. Incubation was carried out at 30 °C for 4 h under limited oxygen conditions. Simulated intestinal fluid tolerance was examined by determining total viable cell counts in intestinal fluid withdrawn at 0, 2, and 4 h by standard plate count method. Meanwhile, all the selected isolates were inoculated into normal pH PBS without pepsin, bile salt, or pancreatin and incubated under the same condition as the simulated GI tract fluids for control. The experiment was performed in three independent replicates. Survival rate was calculated according to the following equation described by Bao et al. [24]:Survival rate (%) = (log CFU/mL N_1_/log CFU/mL N_0_) × 100(1)
where N_1_ represents the total viable count of probiotics after treatment by simulated GI tract juices or normal pH PBS (control) for 7 h, and N_0_ represents the total viable count of probiotics before treatment.

### 2.8. Safety Evaluation of Selected Bacterial and Yeasts Isolates

#### 2.8.1. Hemolytic Activity Test

Selected probiotic cultures grown overnight were streaked on the surface of Blood agar (Tryptic Soy Agar (TSA) (HiMedia) supplemented with 5% (*w*/*v*) sheep blood) [25]. The plates were incubated at 30 °C for 48 h under limited oxygen and aerobic conditions for LAB and yeast, respectively. Blood agar plates were examined for signs of β-hemolysis (clear zones around colony), α-hemolysis (green-hued zones around colonies), or γ-hemolysis (no zones around colonies). *Staphylococcus aureus* ATCC 25923 and *Escherichia coli* ATCC 25922 were used as the positive (β-hemolysis) and negative (γ-hemolysis) control, respectively.

#### 2.8.2. Antibiotics Susceptibility

The antibiotic susceptibility of selected probiotic isolates was determined by the agar disc diffusion method [26]. The following antibiotics were tested in the form of discs: bacitracin (10 µg) (HiMedia), cefpirome (30 µg) (Oxoid), chloramphenicol (30 µg) (Oxoid), clarithromycin (15 µg) (Oxoid), penicillin (10 µg) (Oxoid), sulfamethoxazole/trimethoprim (STX) (25 µg) (Oxoid), tetracycline (30 µg) (Oxoid), and vancomycin (30 µg) (Oxoid). The plates were incubated at 30 °C for 24 h under limited oxygen and aerobic conditions for LAB and yeast, respectively, and diameter of the inhibition zones was measured by using vernier caliper.

### 2.9. Probiotic Bacteria Identification

DNA of selected probiotic bacteria was extracted from cells grown in MRS broth under limited oxygen conditions at 30 °C for 18–24 h and purified according to the method described by Saito and Miura [27] and Tanasupawat et al. [28]. The 16S rDNA was amplified by polymerase chain reaction (PCR) using 27F (5′-AGA GTT TGA TCC TGG CTC A-3′) and 1492R (5′-GGT TAC CTT GTT ACG ACT T-3′) as primers [29]. The 16S rDNA was amplified by Biometra TAdvanced Thermal Cycler (Analytik Jena). Successful amplification of 16S rDNA was validated by visual inspection of PCR product in agarose gel electrophoresis. The separated DNA bands compared to the DNA marker were visualized under UV light using UV Transilluminator (Cleaver). PCR products which showed a positive DNA band of approximately 1.5 kb were sent to Macrogen company for DNA sequencing.

### 2.10. Probiotic Yeast Identification

Selected probiotic yeast isolates were cultured on YM agar under aerobic conditions at 30 °C for 48 h. A single colony of each isolate was selected for DNA extraction. Yeast DNA was extracted by the method previously described by Jutakanoke et al. [30] and Manitis et al. [31]. D1/D2 domain of 26S rDNA was amplified by PCR using F63 (5′-GCA TAT CAA TAA GCG GAG GAA AAG-3′) and LR3 (5′ GGT CCG TGT TTC AAG ACG-3′) as primers [32]. Successful amplification and visualization of DNA bands were performed as described in the identification of probiotic bacteria. PCR products (~500 bp) were sent to Macrogen company for DNA sequencing.

### 2.11. Nucleotide Sequencing and Analysis

DNA sequencing results were aligned using the BioEdit program and connected into a complete sequence. Generated sequences of yeast were compared with 26S rDNA (D1/D2 domain) sequence of related species by nucleotide blast (BLASTn) search (National Center for Biotechnology Information; NCBI). For bacteria, the closest 16S rDNA sequences were identified by blast search using the EzBioCloud server [33]. The highest 16S rDNA or D1/D2 domain of 26S rDNA sequence identity score implied the most closely related species. Phylogenetic tree based on the neighbor joining (NJ) [34] was constructed using MEGA 7.0 software [35]. Gaps and missing data treatments were completely deleted before the calculation. Bootstrap analysis was performed from 1000 random re-samplings [36,37]. The 16S rDNA sequences were submitted to DDBJ/ EMBL/GenBank and were publicly available. The selected microbial strains used in this study were deposited at TISTR culture collection, Pathumthani, Thailand.

### 2.12. Statistical Analysis

All the experiments were performed in triplicate. Homogeneity of variances and normality tests were performed. Data were shown as mean ± standard deviation (SD). Analysis of variance and comparison of the significance of the difference of the mean value of OD_640_ of tested isolates between the bile salt tolerance test group and its control group as well as acid tolerance test group and its control group were performed by independent *t*-test with 95% confidence level using IBM SPSS Statistics version 25. The *p*-values less than 0.05 were considered to be statistically significant.

## 3. Results

### 3.1. Isolation of Probiotic Microbes

A total of 242 microbial isolates from 27 fish GI tract samples (66 bacteria and 176 yeasts) were successfully isolated from Climbing perch (48 bacteria and 114 yeasts), Striped snakehead (8 bacteria and 14 yeasts), Soldier river barb (2 bacteria and 7 yeasts), Common carp (5 bacteria), *L. calcarifer* (10 yeasts), and Nile tilapia (3 bacteria and 31 yeasts).

The morphological and physiological characteristics of all isolates were then further examined. All the bacterial isolates were Gram-positive, non-spore forming, catalase negative, and showed a yellow diffusion halo around the colonies on MRS medium supplemented with bromocresol purple. Based on bacterial morphology and arrangement, they were classified as diplococci, diplobacilli, diplococcobacilli, streptobacilli, streptococcobacilli, and coccobacilli in pairs and clusters. Yeast cells showed six different morphologies, including ovoidal, ovoidal with true mycelium, ovoidal with pseudomycelium, spherical, elongated, and irregular shape.

### 3.2. Pathogenic Bacteria Inhibiting Test

All 66 bacteria isolated from the fish GI tract samples were then tested to determine their inhibitory activity against fish pathogens. Both *A. hydrophila* MUVS 2018 AH 001 and *A. sobria* MUVS 2017 AS growth were inhibited by 65 bacterial isolates. Additionally, *A. veronii* vsmu 083, *E. ictaluri* 2010/12 EI, *E. tarda* MUVS 2018 ET1, and *S. agalactiae* MUVS 2017 SA 001 growth were inhibited by 4, 16, 35, and 36 bacterial isolates, respectively. Of all 66 bacterial isolates, AT8/5 demonstrated outstanding ability in inhibition of all fish pathogenic bacteria. Collectively, most of our 23 isolates were able to inhibit the growth of three pathogenic bacterial strains. The figures of plates showing clear inhibition zones of some probiotic isolates are shown in Appendix A.

Of all 176 yeast isolates, 15 yeasts demonstrated inhibitory activity against the growth of fish pathogenic bacteria. *A. hydrophila* MUVS 2018 AH 001 and *E. tarda* MUVS 2018 ET1 growth were inhibited by one yeast isolate. *E. ictaluri* 2010/12 EI and *S. agalactiae* MUVS 2017 SA 001 growth were inhibited by 18 and 10 yeast isolates, respectively. However, only two pathogenic strains (*A. sobria* MUVS 2017 AS and *A. veronii* vsmu 083) were unable to be inhibited by all yeast isolates. Most isolates (11 isolates) were successfully able to inhibit the growth of at least one pathogenic bacterial strain. Among the tested pathogenic strains, *A. hydrophila* MUVS 2018 AH 001 was more sensitive towards all the probiotic isolates whereas *A. veronii* vsmu 083 showed high resistance towards all the probiotic strains.

### 3.3. Probiotic Properties Tests

Fifty-four microbial isolates consisting of 39 bacterial isolates and 15 yeast isolates with the ability to inhibit pathogenic bacteria were then selected to examine their probiotic properties. All 39 bacterial isolates could inhibit at least three strains of pathogenic bacteria. All yeasts could inhibit at least one strain of pathogenic bacteria. The key characteristics of probiotic properties included the ability to withstand bile salt, acid tolerance, adhesion on the abiotic surface, and biofilm formation.

#### 3.3.1. Acid Tolerance Test

The effect of low pH on bacterial survival was investigated, and the results showed that only nine bacterial isolates could survive at this condition with poor growth (0.1 ≤ OD_640_ < 1) after 24 h of incubation. As expected, all yeasts isolated via a direct survival-selection strategy under extremely acidic media could survive under YM media at pH 2.5.

#### 3.3.2. Bile Salt Tolerance Test

Both bacterial and yeast isolates were tested for bile salt tolerance on specific media supplemented with 1% (*w*/*v*) of bile salt at 30 °C under oxygen limited (for bacteria) and aerobic conditions (for yeast). All bacterial isolates could survive under the tested condition with poor growth (20 isolates) and medium growth (19 isolates) after 24 h of incubation (Table 1 and Appendix A). All yeast isolates demonstrated higher resistance to bile salt when compared to all bacterial isolates. Among all tested yeast isolates, eight (YCS1/1, YCS1/2, YCS1/3, YON3/4, YAT1/6, YAT8/2, YAT10/8, and YAT10/9) outperformed by showing higher growth based on OD_640_ value compared to their control group after 24 h of incubation, with statistically significant differences (*p*-value = 0.001, 0.029, 0.000, 0.019, 0.048, 0.000, 0.007, and 0.000, respectively) (Table 2 and Appendix A).

#### 3.3.3. Adhesion Property Test

Among 54 probiotic isolates, only three isolates of yeast exhibited strong adhesion properties (Abs_590_ ≥ 1) while ten isolates (eight bacteria and two yeasts) exhibited intermediate adhesion properties (0.5 ≤ Abs_590_ < 1). The rest of the isolates (31 bacteria and 10 yeast) demonstrated poor adhesion properties (0.1 ≤ Abs_590_ < 0.5).

#### 3.3.4. Biofilm Formation Test

The biofilm producing colony appeared as a black colony with a black medium around it on CRA, which indicated EPS production. Among 39 bacterial isolates, 34 showed biofilm formation ability while all yeast grew on CRA media with red colonies, which indicated non-biofilm formation ability.

### 3.4. Detection of Antibacterial Factor

Of all 54 isolates selected for determination of probiotic properties, isolates that met the requirement of at least two probiotic criteria based on acid tolerance, bile salt tolerance, adhesion property test, and biofilm formation test were then further investigated for antibacterial factor production and survival rate determination under simulated GI fluid. Based on the requirement for probiotic criteria, three bacterial isolates (CS1/3, CE1/1, and AT8/5) and three yeast isolates (YCS1/1, YCS1/3, and YON3/2) were selected for the detection of antibacterial factor formation (Table 3 and Table 4 and Appendix A).

According to the inhibition assay in the agar slab test, all six isolates showed bacterial inhibition capacity. The antibacterial ability of the six isolates was then further examined by agar well diffusion assay. The non-neutralized CFS of three bacterial isolates produced inhibition zones against fish pathogens with diameters that are not much different from the agar slab method (Table 5 and Appendix A). Nevertheless, when the pH of the probiotic bacterial CFS was neutralized to 7.0 and boiled at 100 °C, no inhibition zone was observed. These results indicate that the inhibition zone was potentially caused by either acidic compounds or heat labile compounds generated by tested isolates because neither non-neutralized nor neutralized CFS of selected probiotic yeast showed inhibition zones against fish pathogens (Table 5 and Appendix A). These finding suggest that yeasts’ inhibitory activity might be potentially triggered when exposed to bacterial pathogens.

### 3.5. Simulated GI Tract Tolerance Test

The effects of gastric and intestinal juices on the survival of selected isolates are shown in Figure 1 and Appendix A. All tested probiotics except CS1/3 and CE1/1 were resistant to the action of simulated gastric and intestinal juices. AT8/5, YCS1/1, YCS1/3, and YON3/2 maintained their viability with insignificant death after 7 h of simulated GI tract exposure. Compared to their control groups, the survival rate for each isolate was 108.35 ± 0.33% (YON3/2), 107.14 ± 0.31% (YCS1/1), 99.49 ± 7.41% (YCS1/3), 76.11 ± 0.75% (AT8/5), and 0% (CS1/3 and CE1/1). Only CS1/3 and CE1/1 presented a reduction in viable cell numbers of approximately five and three log cycles, respectively, after 1.5 h of exposure and died after time of incubation up to 3 h. These results show that these two isolates are sensitive to the simulated GI tract. Remarkably, yeast isolates YCS1/1 and YON3/2 of the tested group showed higher survival rates than their control group after 7 h of incubation, with statistically significant differences (*p*-value = 0.000 and 0.016, respectively) (Figure 2 and Appendix A).

### 3.6. Safety of Selected Probiotic Isolates

Hemolysis activity and antibiotic resistance were performed to determine the safety of probiotic strains. None of the investigated strains showed β-hemolytic activity when grown in blood agar with sheep blood. All bacterial isolates demonstrated α-hemolysis while all yeast isolates showed γ-hemolysis. 

The antibiotic susceptibility of selected isolates was investigated by using eight antibiotics (Table 6). All bacterial isolates (except AT8/5) were resistant to cefpirome, STX, vancomycin, and penicillin but susceptible to bacitracin, chloramphenicol, clarithromycin, and tetracycline. In the case of yeasts, the isolates YCS1/3 and YON3/2 showed resistance to all antibiotics used, whereas YCS1/1 demonstrated resistance to only cefirome.

### 3.7. Genotypic Identification of Selected Probiotics

The six selected isolates were identified by 16S rDNA and the D1/D2 domain of 26S rDNA sequence analysis for bacterial and yeast identification, respectively. The 16S rDNA sequence analysis of the isolate CE1/1 (Accession number LC735573) and CS1/3 (LC735574) showed the highest similarity to Weissella paramesenteroides (ACKU01000017) with 99.86 and 99.79% identity, respectively. The isolate AT8/5 (LC735575) showed the highest similarity to Lactiplantibacillus argentoratensis (CP032751) with 99.79% identity. The D1/D2 domain sequence analysis of the isolate YCS1/1 (LC735679) and YCS1/3 (LC735680) showed 100% similarity with Kodamaea ohmeri (MN268772). The isolate YON3/2 (LC735681) showed 100% similarity with Candida tropicalis (LC496591). The GenBank accession numbers of each identified probiotic isolate were presented in Appendix A. The phylogenetic trees constructed by the neighbor joining method based on 16S rDNA and D1/D2 domain sequences showing the position of each isolate are indicated in Figure 3 and Figure 4, respectively.

## 4. Discussion

On the basis of the host-specificity exhibited by members of autochthonous microbes and homologous hosts, we hypothesized that microbes readily isolated from fish GI tracts could be potentially applied as probiotic microbes to prevent pathogenic bacterial infection in fish farming. In this experiment, 242 microbial isolates, 66 bacteria and 176 yeasts were successfully isolated from fish GI tract samples from seven species, including Climbing perch, Striped snakehead, Common carp, Asian sea bass, Nile tilapia, and Soldier river barb. All the bacterial isolates were Gram-positive, non-spore forming, and catalase negative and were identified as LAB based on morphological and physiological characterization [38]. Our yeast isolates were screened using a direct survival selection strategy under an extremely acidic environment. They were identified as acid tolerant probiotic yeasts that can be found in the GI tract of animals.

Antibacterial activity against pathogens is one of the key functional requirements for probiotic strain selection. Our results showed that 66 bacterial isolates and 15 yeast isolates demonstrated inhibitory effects against fish pathogenic bacteria. Previous studies also reported that *Lactococcus lactis* [10] and yeasts [12,13] isolated from the intestine of fish showed inhibitory effects against pathogens. Many reports indicated that the antibacterial effect manifested by LAB is due to the production of some antimicrobial compounds (AMCs) such as organic acids (lactic acid, acetic acid, etc.), hydrogen peroxide (H_2_O_2_), carbon dioxide (CO_2_), diacetyl (2,3-butanedione), and bacteriocins [39]. LAB produces organic acids as metabolites decreasing the intracellular pH which interrupts DNA and protein functions, leading to cell growth inhibition and eventually cell death. Moreover, LAB possesses the ability to produce either heat-stable or heat-sensitive proteins with antibacterial properties called bacteriocins [40] that provoke the development of pores in the cell membrane of susceptible bacteria through which electrolytes in the cytoplasm leak out, thus leading to their death [41]. In this study, both non-neutralized CFS and neutralized and boiled CFS were investigated on their antibacterial property. The non-neutralized CFS of LAB isolates demonstrated antimicrobial ability in an agar well diffusion assay. However, no inhibition zone was observed when the pH of the CFS was neutralized to 7.0 and subsequently boiled. This result indicated that the inhibitory activity was attributed to either secreted organic acids or possibly heat-sensitive bacteriocin. The antagonistic activity of the yeast isolates against the fish pathogenic bacteria might be attributed to the production of certain yeast products i.e., ethanol and antimicrobial compounds such as killer toxins or mycocins [41]. In the present work, the results of the agar slab method showed that yeast isolates demonstrated growth-inhibiting properties against fish bacterial pathogens, but neither non-neutralized nor neutralized and boiled CFS exhibited antagonistic activity in an agar well diffusion assay. This might be because the antagonistic activity of the CFS was extremely weak or absent. The release of AMCs by microbes is influenced by many factors such as culture conditions, cell density, and population kinetics [42]. In parallel to our study, Polak-Berecka et al. [43] also suggest that the most potent antimicrobial activity was observed when the live cells of probiotics were used. Thus, the growth inhibition of fish pathogens was attributed to the live culture of yeast, not by yeast CFS.

The successful probiotic candidates with antagonistic effects against pathogens should tolerate high bile salt concentration and acidic pH in the fish GI tract. The common stomach pH value in Nile tilapia is very low (approximately 2.5) because of the presence of hydrochloric acid (HCl) activating the action of pepsin. In the present study, 39 probiotic bacteria isolates were selected to investigate these properties. In this research, the probiotic yeast isolates were isolated under an extremely acidic culture condition at pH 2.5; therefore, all yeast isolates were acid tolerant by default. Nine selected probiotic bacteria could survive at pH value 2.5 for 24 h. Many studies reported that potential probiotics such as *Lactobacillus* spp. and *Enterococcus* spp. isolated from freshwater fish could survive at pH 2 [44,45]. The most important pH homeostasis mechanism of LAB and some yeast species is the maintenance of intracellular pH by pumping protons (H^+^) that result from extracellular acid dissociation processes via ATP hydrolysis using proton-translocating ATPase (H^+^-ATPase) [46,47]. In addition, bile salt has strong toxic effects on the cell membrane of microbes [48]. The concentration of bile salt in the fish GI tract ranged from 0.4 to 1.3% [44]. The bacterial probiotics in this study demonstrated bile salt tolerance in the presence of 1% bile salt for 24 h, with poor (20 bacterial isolates) and medium growth (19 bacterial isolates). Under the same tested condition for bile salt tolerant assay, the 15 yeast isolates showed stronger tolerance against the bile salt than bacterial isolates. Our results are in agreement with previous studies reporting bile salt tolerance of many fish gut-isolated microbes [10,49]. The tolerance to bile salt of the LAB and some yeast strains is presumably related to the capability to create the bile salt hydrolase (BSH) enzyme that protects them against the toxicity of bile salts by hydrolysis of amide bond in bile salt molecules [48]. Nevertheless, the valid mechanism of BSH of yeast has yet to be elucidated [50].

Another important probiotic property is the ability to adhere to the GI tract mucous membrane, promoting the persistence of probiotics over long periods of time. The results of the probiotic adhesion assessment were obtained by absorbance of crystal violet dissolved from the cells attaching to the walls of the test tube. Among 54 selected probiotic isolates, 15 isolates of yeast and 39 isolates of bacteria exhibited adhesion property. Several studies showed the ability of probiotics to attach to the hydrophobic surface [13,51]. The probiotic adhesion depends on the cell surface hydrophobicity (CSH) affecting binding of probiotics to either biotic or abiotic hydrophobic surfaces e.g., glass, medical devices, and epithelial cells [52]. For this reason, probiotics with adhesion to the glass surface might be able to adhere and colonize to the hydrophobic hydrocarbon of host intestinal epithelial cells. Furthermore, adhesion ability is a critical factor in allowing probiotics to localize and form biofilms rendering their colonization and safeguarding from unfavorable conditions of GI tract, which in turn limits the adherence and colonization of pathogenic strains on the intestinal mucous membrane [53]. Biofilms are normally composed of microbial cells and secreted exopolysaccharide (EPS), which is essential for maintaining the stability of the biofilm structure. EPS produced by microbes has been detected by a change of colony color on CRA.

The six selected isolates which met the probiotic criteria were then subjected to the detection of survival rate under simulated GI fluid. To qualify as probiotics used in Nile tilapia, bacterial or yeast strains must survive for approximately 7 h in the gastric conditions of the stomach and reach the intestine alive in which they will exert their function because the estimated total digestion time of Nile tilapia is 7.15 h [54]. Some selected probiotic bacteria and yeasts showed strong tolerance in gastric and intestinal conditions. These results are also in agreement with viability reported for other probiotics such as *Lactobacillus* spp. [55] and *Candida* sp. [56] in the presence of a simulated GI tract. HCl and digestive enzymes including pepsin and pancreatin in the GI tract can inhibit most microbes. With respect to resistance to pepsin, probiotic strains may convert the content of amino acids of their membrane proteins into non-hydrophobic amino acid which is not selectively hydrolyzed by the action of pepsin [57]. Many researchers have reported an insignificant effect of pancreatic enzymes on intestinal microbes compared to bile salt [58]. Notably, our yeast isolates (YCS1/1 and YON3/2) showed significantly higher survival rates than their control group after 7 h of incubation. These results indicate that bile salt possibly contributes to the growth of the yeasts by serving as a carbon and nitrogen source. Hernández-Gómez et al. [50] reported that glycine or taurine amino acid released from conjugated bile salt when the amide bonds earn hydrolysis by the BSH could be used as carbon and nitrogen sources for growth by the intestinal microbiota.

According to safety considerations, hemolysis activity and antibiotic resistance were used to evaluate the safety of probiotic strains. The absence of β-hemolytic activity is regarded as a safety prerequisite for selecting probiotic strains [59]. While the result of β-hemolysis is considered deleterious, γ-hemolysis and α-hemolysis are regarded to be safe [60,61]. In the current study, γ-hemolysis was shown by all selected yeast isolates, whereas α-hemolysis was the resultant acquired for all bacterial isolates. Diguță et al. [62] also reported that all yeast strains with γ-hemolysis possessed valuable probiotic traits for aquaculture use. Similar observations of α-hemolytic probiotics were also reported in some LAB strains [55,63,64].

To be regarded as having probiotic properties, a probiotic must not harbor acquired and transferable antibiotic-resistance genes [65]. In this study, all of the bacterial isolates showed resistance to certain antibiotics used. As reported by Curragh and Collins [66], antibiotic-resistant probiotics might have negative consequences because antibiotic-resistance genes might be transferable to pathogenic bacteria. Notwithstanding, the inherent resistance of LAB strains to some antibiotics was encoded by antibiotic resistance genes contained in chromosomes, and thus the transmission of antibiotic resistance would be much lower compared to ones in plasmid [67,68]. However, the beneficial effect of antibiotic-resistant bacterial probiotics is that probiotics can be co-administered with therapeutic antibiotics for treatment of an infectious disease [69]. In our study, all yeast isolates (except YCS1/1) displayed resistance to all antibiotics. The resistance of the yeasts to antibiotics also enables them to be suitable for therapeutic application in patients undergoing antibiotic treatment [62,70].

The six selected isolates, namely CS1/3, CE1/1, AT8/5, YCS1/1, YCS1/3, and YON3/2, showed the highest similarity to *W. paramesenteroides*, *W. paramesenteroides*, *L. argentoratensis*, *K. ohmeri*, *K. ohmeri*, and *C. tropicalis*, respectively. Recently, studies of probiotic properties of *W. paramesenteroides* isolated from fish such as Arapaima (*Arapaima gigas*) [71] were reported. However, it has not been formerly reported from Striped snakehead or Soldier river barb. To the best of our knowledge, this is the first study to report the isolation of culturable probiotic *W. paramesenteroides* that was effective against several Nile tilapia pathogens from Striped snakehead and Soldier river barb. *L. argentoratensis* comb. nov. was re-evaluated from *Lactobacillus argentoratensis* or *Lactobacillus plantarum* subsp. *argentoratensis* by a combination of 16S rDNA sequence analysis and phylogenomic treeing [72]. The characterization of probiotic properties of *L. argentoratensis* and *L. plantarum* subsp. *argentoratensis* was rarely reported [73]. Furthermore, the isolation of *L. argentoratensis* from fish has not been previously reported. Accordingly, this is the first report of *L. argentoratensis* isolated from Climbing perch which has exhibited probiotic properties and antibacterial activity against Nile tilapia pathogens. Even though yeast strains have also been reported to possess probiotic ability, few reports regarding probiotic yeasts are available. In the present study, two species of yeast, *K. ohmeri* and *C. tropicalis,* were found to be probiotics. The probiotic potential of these yeast strains was previously found [74,75,76]. Although the isolation of these yeasts from several marine fish was reported [77,78], there is no record of isolation of these yeast strains from freshwater fish. Consequently, this study could potentially be the first record demonstrating antimicrobial activity against Nile tilapia pathogens and probiotic characteristics of *K. ohmeri* and *C. tropicalis* isolated from the GI tract of Striped snakehead and Nile tilapia, respectively. Collectively, probiotic bacteria AT8/5 (*L. argentoratensis*) demonstrated an outstanding ability to inhibit all tested bacterial pathogens. Additionally, probiotic yeast YON3/2 (*C. tropicalis*) can inhibit two species of tested pathogenic bacteria. Our isolated probiotic bacteria and yeast could be promising probiotics for application in Nile tilapia farming as feed additives to prevent bacterial infection.

## 5. Conclusions

In this study, the isolation of probiotics from fish and evaluation of their inhibitory activities against Nile tilapia’s pathogenic bacteria as well as other properties essential for probiotic functions were investigated. The selected isolates were identified by 16S rDNA and the D1/D2 domain of 26S rDNA sequence analysis for bacterial and yeast identification, respectively. These six isolates, namely CS1/3, CE1/1, AT8/5, YCS1/1, YCS1/3, and YON3/2, showed the highest similarity (%Identity) to *W. paramesenteroides, W. paramesenteroides, L. argentoratensis, K. ohmeri, K. ohmeri*, and *C. tropicalis*, respectively. *L. argentoratensis* (AT8/5) and *C. tropicalis* (YON3/2) could be potentially employed as probiotics in Nile tilapia farming as feed additives to prevent the infection of pathogenic bacteria in Nile tilapia or possibly other economic fish. The use of probiotics could be used as a replacement for conventional antibiotic treatment, thus lowering public health risk, in particular, antibiotic resistance in Nile tilapia and humans. Although promising, further assessments, including next-generation sequencing (NGS) and in vivo studies such as cytotoxicity, colonization ability, stimulation of growth and immune system, production of adverse effects, and the prevention of infectious disease in Nile tilapia models are required to confirm their suitability as probiotics for Nile tilapia.

## Figures and Tables

**Figure 1 vetsci-10-00129-f001:**
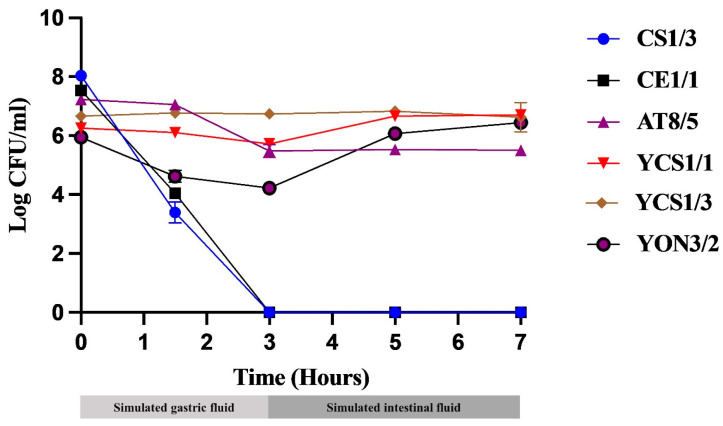
Testing probiotic tolerance in simulated GI tract conditions. Bacteria and yeasts isolated from fish GI tracts were determined for their probiotic activity by testing acid and bile tolerance in 3 h of simulated gastric fluid (SGF) and, subsequently, 3 h in simulated intestinal fluid (SIF), respectively. Enumeration of each isolate was performed by serial plating on MRS and YM agar for bacteria and yeast, respectively under limited oxygen conditions.

**Figure 2 vetsci-10-00129-f002:**
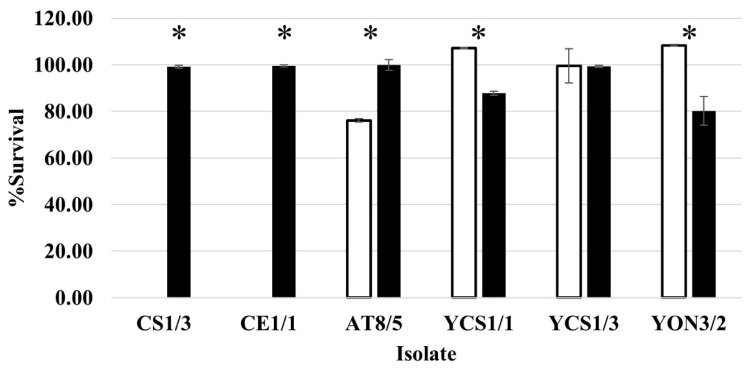
Percentage of survival of individually selected probiotics at simulated GI tract for 7 h. White bars: tested group; black bars: control group; *: statistically significant difference (*p* < 0.05) compared with their control.

**Figure 3 vetsci-10-00129-f003:**
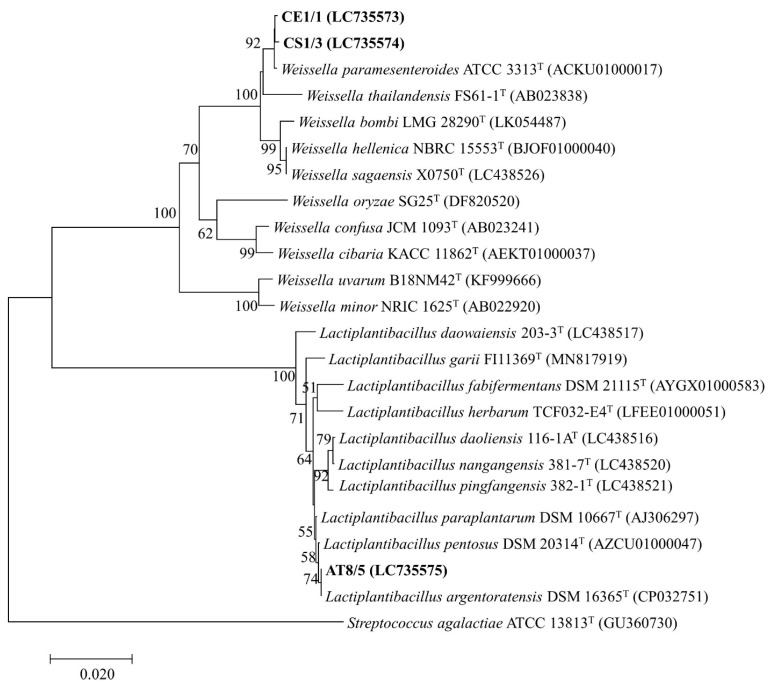
Phylogenetic tree constructed by the neighbor joining method based on 16S rDNA sequences indicating position of probiotic bacterial isolates. *Streptococcus agalactiae* ATCC 13813^T^ was used as an outgroup. Numbers at the nodes indicate bootstrap values based on 1000 replicates.

**Figure 4 vetsci-10-00129-f004:**
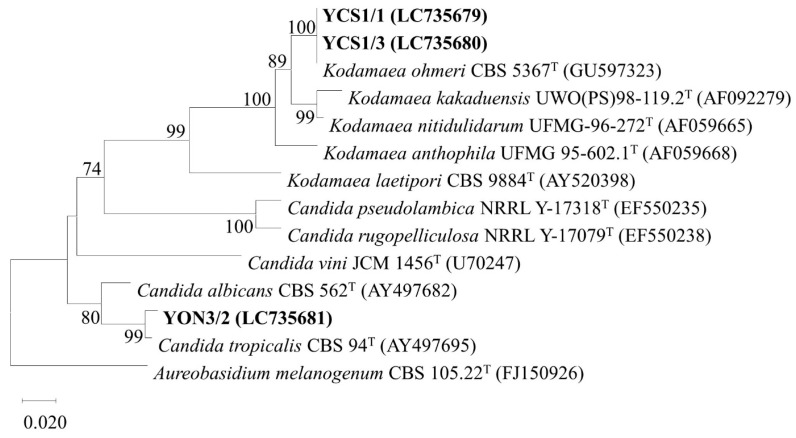
Phylogenetic tree based on neighbor joining analysis of D1/D2 domain sequences showing the relationship of probiotic yeast isolates. *Aureobasidium melanogenum* CBS 105.22^T^ was used as an outgroup. Numbers at the nodes indicate bootstrap values based on 1000 replicates.

**Table 1 vetsci-10-00129-t001:** Effect of bile salt on the growth of some bacterial isolates. The uppercase letters show the comparison of the OD_640_ value of each isolate in the tested and control group at an incubation period of 6 h; the lowercase letters show the comparison of the OD_640_ value of each isolate in the tested and control group at 24 h of incubation; different letters in each column show a statistically significant difference.

Isolates	OD_640_			
	Control Group		Tested Group	
	6 h	24 h	6 h	24 h
CS1/3	1.86 ^A^	3.23 ^a^	0.39 ^B^	1.42 ^b^
CE1/1	1.51 ^A^	2.99 ^a^	0.45 ^B^	1.39 ^b^
AT8/5	1.19 ^A^	1.90 ^a^	0.39 ^B^	0.67 ^b^

**Table 2 vetsci-10-00129-t002:** Effect of bile salt on the growth of some yeast isolates. The uppercase letters show the comparison of the OD_640_ value of each isolate in the tested and control group at an incubation period of 6 h; the lowercase letters show the comparison of the OD_640_ value of each isolate in the tested and control group at 24 h of incubation; different letters in each column show a statistically significant difference.

Isolates	OD_640_			
	Control Group		Tested Group	
	6 h	24 h	6 h	24 h
YCS1/1	0.39 ^A^	2.89 ^a^	0.21 ^B^	5.55 ^b^
YCS1/2	0.43 ^A^	3.15 ^a^	0.20 ^B^	4.67 ^b^
YCS1/3	0.39 ^A^	2.74 ^a^	0.36 ^A^	6.20 ^b^
YON3/4	1.24 ^A^	4.45 ^a^	0.89 ^B^	5.07 ^b^
YAT1/6	0.71 ^A^	6.25 ^a^	0.35 ^B^	6.64 ^b^
YAT8/2	0.95 ^A^	2.75 ^a^	0.44 ^B^	4.58 ^b^
YAT10/8	0.93 ^A^	2.52 ^a^	0.93 ^A^	4.43 ^b^
YAT10/9	1.09 ^A^	2.83 ^a^	0.99 ^A^	4.34 ^b^

**Table 3 vetsci-10-00129-t003:** Antibacterial properties of selected isolates by using agar slab assay (Mean ± SD).

Isolates	Inhibition Zone (mm) of Pathogenic Bacteria					
	AH	AS	AV	EI	ET	SA
Positive control	26.00 ± 0.00	28.5 ± 0.50	26.17 ± 0.29	30.17 ± 0.29	33.00 ± 1.00	36.00 ± 1.41
Negative control	-	-	-	-	-	-
CS1/3	9.25 ± 0.25	9.42 ± 0.14	-	-	12.17 ± 0.29	9.42 ± 0.14
CE1/1	8.67 ± 0.29	8.75 ± 0.35	-	-	10.33 ± 0.29	8.33 ± 0.29
AT8/5	13.50 ± 0.71	22.83 ± 1.15	7.17 ± 0.29	16.33 ± 0.58	13.17 ± 0.29	9.83 ± 0.76
YCS1/1	-	-	-	-	-	7.00 ± 0.00
YCS1/3	-	-	-	-	-	7.08 ± 0.14
YON3/2	-	-	-	7.17 ± 0.29	-	7.25 ± 0.43

AH: *A. hydrophila* MUVS 2018 AH 001; AS: *A. sobria* MUVS 2017 AS; AV: *A. veronii* vsmu 083; EI: *E. ictaluri* 2010/12 EI; ET: *E. tarda* MUVS 2018 ET1; SA: *S. agalactiae* MUVS 2017 SA 001; -: No inhibition zone.

**Table 4 vetsci-10-00129-t004:** Probiotic properties of selected isolates.

Probiotic Properties	Probiotic Bacteria			Probiotic Yeast		
	CS1/3	CE1/1	AT8/5	YCS1/1	YCS1/3	YON3/2
Acid tolerance at 24 h	-	+	-	+++	+++	+++
Bile salt tolerance at 24 h	++	++	+	+++	+++	+++
Adhesion property	*	**	*	***	***	*
Biofilm formation	-	-	✓	-	-	-

+: poor tolerance; ++: medium tolerance; +++: great tolerance; *: poor adhesion; **: medium adhesion; ***: great adhesion; ✓: biofilm formation.

**Table 5 vetsci-10-00129-t005:** Determination of the inhibitory factors of selected probiotic isolates against fish pathogens, as determined by agar well diffusion methods (Mean ± SD).

Isolates		Inhibition Zone (mm) for Fish Pathogenic Bacteria					
		AH	AS	AV	EI	ET	SA
CS1/3	C	7.17 ± 0.29	15.00 ± 1.00	X	X	7.17 ± 0.29	6.83 ± 0.29
	N	-	-	X	X	-	-
CE1/1	C	8.67 ± 0.29	13.83 ± 0.76	X	X	10.33 ± 0.29	9.00 ± 0.50
	N	-	-	X	X	-	-
AT8/5	C	10.33 ± 0.58	22.83 ± 1.15	7.00 ± 0.00	16.00 ± 0.00	11.67 ± 0.29	9.83 ± 0.29
	N	-	-	-	-	-	-
YCS1/1	C	X	X	X	X	X	-
	N	X	X	X	X	X	-
YCS1/3	C	X	X	X	X	X	-
	N	X	X	X	X	X	-
YON3/2	C	X	X	X	-	X	-
	N	X	X	X	-	X	-

AH: *A. hydrophila* MUVS 2018 AH 001; AS: *A. sobria* MUVS 2017 AS; AV: *A. veronii* vsmu 083; EI: *E. ictaluri* 2010/12 EI; ET: *E. tarda* MUVS 2018 ET1; SA: *S. agalactiae* MUVS 2017 SA 001; C: Non-neutralized CFS; N: Neutralized CFS; -: No inhibition zone; X: Not determined because these isolates demonstrated no inhibition zone based on the previous experiment (agar slab method) shown in Table 3.

**Table 6 vetsci-10-00129-t006:** Antibiotic resistance of the selected isolates.

Antibiotics	Isolates					
	CS1/3	CE1/1	AT8/5	YCS1/1	YCS1/3	YON3/2
Bacitracin	S	S	S	S	R	R
Cefpirome	R	R	R	R	R	R
Chloramphenicol	S	S	S	S	R	R
Clarithromycin	S	S	S	S	R	R
Penicillin	R	R	S	S	R	R
STX	R	R	R	S	R	R
Tetracycline	S	S	S	S	R	R
Vancomycin	R	R	R	S	R	R

S: susceptible; R: resistant.

## Data Availability

Not applicable.

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
