# Peer review of "Lactiplantibacillus argentoratensis and Candida tropicalis Isolated from the Gastrointestinal Tract of Fish Exhibited Inhibitory Effects against Pathogenic Bacteria of Nile Tilapia"

_vetsci, 2023, doi:10.3390/vetsci10020129_

Round 1
Reviewer 1 Report
This is a special study with regard to the probiotics of bacteria and fungi from fish gastro-intestine. The results seem interesting and I look forward to see more experiments to test if this procedure may be used to avoid/decrease diseases of fish. The results are well presented and the manuscript is well written with just some minor suggestions.
Minor
Line 41: that; change to: on
L 58: the import antibiotic-treated fish products; change to: the import of antibiotic-treated fish products
L 83: There have not much been reports on; change to: There have been few reports
L 161: in the present of bile salt; change to: in the presence of bile salt
L 203: was divided into 2 group; change to: was divided into 2 groups
L 218: 1 ml of density adjusted cultures were; change to: 1 ml of density adjusted cultures was
L 246: valicated; change to: validated
L 301+302: in inhibition the growth; change to: in inhibiting
L 311: A. hydrophila…..were more; change to: A. hydrophila…..was more
L 351+352: bile salt possibly contributes the growth of the yeasts; change to: bile salt possibly contributes to the growth of the yeasts
Reviewer 2 Report
Dear authors,
Congratulations on your interesting manuscript. The manuscript is interesting and well structured, describing new potential probiotic candidates that can be applied in Nile tilapia and other freshwater fish productions as preventive agent for certain bacterial diseases.
Nevertheless, you should be aware of the following suggestions/corrections that need to be made:
All manuscript
- The manuscript needs extensive English editing in order to be readable, especially in Abstract, Introduction and Material and Methods section;
- You must be consistent when referring to organisms species in the manuscript. All over the text, for example, you refer to Nile tilapia as "Nile tilapia" or "O. nicotilus" after citing them for the first time the specie. Please select one and keep it uniform.
The same should happen when you cite a specie for the first time. Every time as possible please use a common name and specie name (ex. Nile tilapia (Oreochromis niloticus), or climbing perch (Anabas testudineus));
- Please refer the brands of all the mediums and devices used in the experiment.
Abstract
- An extensive test editing should be performed to improve this section readability (please check suggestions in the manuscript);
- You cannot use as Keywords part of the manuscipt title (ex. Nile tilapia; Pathogenic bacteria). Please correct this.
Introduction
Overall this section is well-structured and contains all the necessary information. However you should adress the following issues:
- An extensive test editing should be performed to improve this section readability (please check suggestions in the manuscript);
- Please correct the homogeneity of citation of species name along this section.
Material and Methods
This section is well-structured and contains all the necessary information. Neverthless, there are some questions that should be specified in the manuscript:
1 - The fish GI tracts were sampled from dead fish, or the fishes were sacrificed before sampling?
2 - How did you preserve the samples for transportation?
3 - In the Statistical Analysis, did you perform any nomality and homogeneity of variance tests?
4 - I would also suggest that some post-hoc tests should be performed to increase the robustness of your statistical analysis.
You should adress other suggestions that are in the attached manuscript, to increase the manuscript readability.
Results
This section is well-structured and contains all the necessary information. I only have some minor suggestions, that are described in the attached manuscript.
Discussion
This section is well written and contains all the necessary information.
Please perform a text editing to improve the manuscript readability. I only have some other minor suggestions, that are described in the attached manuscript.
Conclusion
Please rewrite this section, that should only contain the selected bacterial isolates and future perspectives of the work.
Best regards,
Reviewer 3 Report
It is difficult to publish papers based on these simple in vitro experiments anymore. Recently, various analysis methods including next-generation sequencing have been developed and are used in research. Authors should redesign and revise this study in comparison with the latest research methods and contents.
Safety verification is essential for use as probiotics. Hemolysis, cytotoxicity, presence of toxic genes of isolated probiotics should be investigated. Even if it is isolated from the host's intestines, safety must be verified through feeding experiments. Authors can add figures of plates showing clear inhibition zones in the supplementary file. What kind of bacteriocins specifically affect the antibacterial activity? Does it work for both gram positive and negative? What is the antibiotic resistance of the isolated probiotic candidate strains? Is it safe for antibiotic resistance gene transfer?Author Response
Please see the attachment.

Round 2
Reviewer 2 Report
Dear authors.
Congratulation on the manuscript improvement. However, you must still address some more issues before the manuscript is ready for publication. Your major issues are:
Abstract - Is confusing and hard to read;
Introduction - It lacks a clearly described objective for the work described in the manuscript. It also has some writing problems that should be addressed;
Material and Methods - There are some small imprecisions in the technique's description. It also has some writing problems that should be addressed;
Results - It has some writing problems that should be addressed, to improve the manuscript's clearness and readability;
Discussion - It has some writing problems and other issues that should be addressed, to improve the manuscript's clearness and readability;
Conclusion - Needs to be more directed. It also has some writing problems that should be addressed.
You can check all these suggestions/corrections in the attached file.
Best regards,
